# Emittance preservation in a plasma-wakefield accelerator

C. A. Lindstrøm ®[1,2] ✉, J. Beinortaitė[1,3], J. Björklund Svensson ®[1], L. Boulton[1,4,5], J. Chappell ®[3], S. Diederichs ®[1,6], B. Foster[7], J. M. Garland[1], P. González Caminal ®[1,6], G. Loisch ®[1], F. Peña ®[1,6], S. Schröder ®[1], M. Thévenet ®[1], S. Wesch ®[1], M. Wing ®[1,3], J. C. Wood ®[1], R. D'Arcy ®[1] & J. Osterhoff ®[1]

Radio-frequency particle accelerators are engines of discovery, powering high-energy physics and photon science, but are also large and expensive due to their limited accelerating fields. Plasma-wakefield accelerators (PWFAs) provide orders-of-magnitude stronger fields in the charge-density wave behind a particle bunch travelling in a plasma, promising particle accelerators of greatly reduced size and cost. However, PWFAs can easily degrade the beam quality of the bunches they accelerate. Emittance, which determines how tightly beams can be focused, is a critical beam quality in for instance colliders and free-electron lasers, but is particularly prone to degradation. We demonstrate, for the first time, emittance preservation in a high-gradient and high-efficiency PWFA while simultaneously preserving charge and energy spread. This establishes that PWFAs can accelerate without degradation—an essential step toward energy boosters in photon science and multistage facilities for compact high-energy particle colliders.

In a conventional radio-frequency (RF) particle accelerator, the accelerating field is limited to approximately 100 MV m$^{-1}$ by breakdowns in the metallic accelerator cavities. Consequently, X-ray free-electron lasers[1,2] (FELs) with energy of order 10 GeV, used in photon-science research, are long and expensive. This is even more so for linear electron–positron colliders at the TeV scale[3,4]. By exchanging the accelerating medium from a metal to a plasma, which is not limited by breakdown effects, plasma-based accelerators can provide accelerating fields as high as 100 GV m$^{-1}$, 1000 times larger than RF accelerators. In principle, this promises to make accelerators significantly shorter and cheaper.

Plasma-based acceleration[5–8] can occur when an intense laser pulse or charged-particle beam (known as a driver) traverses a plasma, expelling plasma electrons in its path and driving a charge-density wave behind it—a so-called plasma wake. The resulting separation of electrons and ions creates strong electromagnetic fields, or plasma wakefields, which can be used both to accelerate and focus a trailing particle bunch. In beam-driven plasma-wakefield accelerators (PWFAs), experiments have already demonstrated large energy gain[9,10], high energy-transfer efficiency from the driver to the trailing bunch[11], acceleration across multi-metre-scale accelerator stages[12], as well as potential for high repetition rate[13].

Excellent beam quality is also required for many applications. This includes high charge, short bunch length, low-energy spread, and low emittance—all different facets of a high charge density in phase space, also known as beam brightness. In particular, emittance, which determines how tightly a beam can be focused, strongly affects the performance of FELs and linear colliders. Typically, FELs demand 100 pC-scale bunches of sub-100 fs duration with 0.1% energy spreads and emittances of order 1 mm mrad. Linear colliders, on the other hand,

[1]Deutsches Elektronen-Synchrotron DESY, Hamburg, Germany. [2]Department of Physics, University of Oslo, Oslo, Norway. [3]Department, University College London, London, UK. [4]Department of Physics, Scottish Universities Physics Alliance, University of Strathclyde, Glasgow, UK. [5]The Cockcroft Institute, Daresbury, UK. [6]Universität Hamburg, Hamburg, Germany. [7]John Adams Institute, Department of Physics, University of Oxford, Oxford, UK. ✉e-mail: c.a.lindstrom@fys.uio.no

require nC-scale charge, sub-1% energy spread, and emittances as low as 0.01 mm mrad. For plasma accelerators to be compact, more affordable alternatives to RF accelerators, each stage must not only accelerate with high gradient, efficiency, and repetition rate but also preserve these beam qualities.

Recent experiments have demonstrated that both charge and energy spread can be preserved in a PWFA[14–16], and that a sufficient beam brightness can be maintained during a small energy boost while still allowing FEL gain at infrared wavelengths to occur[17]. However, preservation of emittance has not been established until now.

A beam's root-mean-square (rms) normalized emittance[18], $\epsilon_n$, represents the area of its rms ellipse in transverse phase space, given by $\epsilon_n^2 = \langle x^2 \rangle \langle u_x^2 \rangle - \langle x u_x \rangle^2$, where $x$ is the offset from the nominal trajectory, and $u_x$ is the transverse momentum normalized by the particle mass and the speed of light in vacuum. This quantity is preserved during both acceleration and beam focusing, provided the focusing field is linear (i.e., proportional to the transverse offset), as is the case in ideal quadrupole magnets. Similarly, in the uniform ion channel of a nonlinear plasma accelerator operating in the blowout regime[19,20], the focusing field is also linear, and thus the emittance of an accelerating electron bunch can, in principle, be preserved.

However, many sources of emittance growth can complicate this picture[21]. Firstly, a bunch externally injected into a plasma-accelerator stage must be tightly focused to fit within the 10–100 μm-scale plasma cavity, and its beta function[22] (i.e., the Rayleigh range for a focused particle beam) must be precisely matched to the strong focusing forces therein to prevent an oscillation of the beam size[23]. Any mismatch causes phase mixing in bunches with finite energy spreads[24] and can lead to the sampling of the nonlinear focusing fields near the edge of the cavity, both of which increase emittance. Similar effects occur if the bunch is transversely misaligned[25,26]. The wakefield driver can also indirectly cause emittance growth; in certain cases, particle-beam drivers can develop hose instability[27], which leads to rapid fluctuation of the fields experienced by the trailing bunch. In addition, if the beam driver has sufficient charge density, it can move ions towards the axis, forming an ion-density spike with highly nonlinear focusing fields[28,29]. Lastly, Coulomb collisions between beam and gas or plasma particles can increase the emittance through scattering[30,31]. To avoid emittance growth, all these effects must be evaluated and, if necessary, mitigated.

Not only is it challenging to preserve emittance, but it is also non-trivial both to locate the ideal operating point and to measure accurately the emittance and energy spectrum. In practice, this difficulty scales with energy gain, because the ideal operating region shrinks and the larger (absolute) energy jitter of the accelerated bunch increases the difficulty of making accurate multi-shot emittance measurements. An initial demonstration of emittance preservation is, therefore, best carried out in a plasma accelerator that is long enough to display the relevant sources of emittance growth and be sensitive to the required tuning precision, but short enough to be compatible with current state-of-the-art stability in electron-beam and plasma generation.

In this work, we demonstrate the preservation of emittance in a beam-driven plasma-accelerator stage while simultaneously preserving charge and energy spread. This was accomplished at the FLASH-Forward plasma-accelerator facility[32] at DESY, employing stable and high-quality beams from the FEL facility FLASH[33].

## Results

### Experimental setup

Electron bunches from a photocathode source were accelerated to 1050 MeV by superconducting RF cavities, compressed in two magnetic chicanes, and linearized in longitudinal phase space by a third-harmonic cavity. Active-feedback systems were used to stabilize the charge, energy, orbit, and bunch length. Two bunches were created in a horizontally dispersive section using a three-component mask[34]: two block collimators to remove the head and tail of the bunch, and a notch collimator to split it into a driver- and trailing-bunch pair (see the "Methods" section). Downstream quadrupole and sextupole magnets in a region of large dispersion were then adjusted to align the two bunches transversely. In the subsequent straight section, nine quadrupole magnets (see Supplementary Fig. 1) were used to focus the beam strongly at the entrance of a 50 mm-long capillary filled with argon gas (see Fig. 1a), around which two beam-position monitors (BPMs) measured the beam trajectory. The beam arrived 9.68 μs after a high-voltage discharge ionized the argon, resulting in a central plasma density of ~1.2 × 10^16 cm^{-3} with upstream and downstream density ramps (see "Methods" and Supplementary Fig. 4).

The main diagnostics, downstream of the plasma-accelerator stage, were two electron energy spectrometers based on 1 m-long

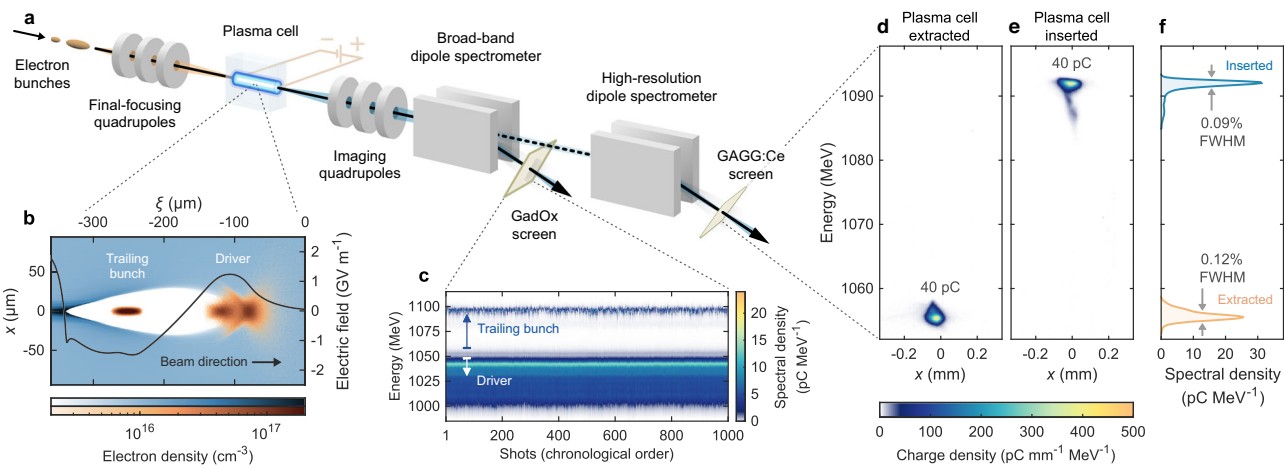

**Fig. 1 | Setup. a** Two electron bunches were focused by quadrupole magnets into a plasma created by a high-voltage discharge, then captured and imaged with another set of quadrupoles onto one of two dipole spectrometers. **b** A PIC simulation with plasma (blue colour scale) and beam electrons (orange colour scale) shows the leading driver bunch creating a wake in which a trailing bunch experiences GV m^{-1} on-axis accelerating fields (black line) and strong transverse focusing; $x$ and $\xi = z - ct$ denote the directions perpendicular and parallel to the direction of motion, respectively. **c** The resulting energy spectrum, measured by a broad-band spectrometer, shows that the driver loses energy (white arrow) and the trailing bunch gains energy (blue arrow), with high stability. **d–f** Representative shots on a downstream high-resolution spectrometer show that the trailing bunch had consistent charge before (**d**) and after acceleration (**e**), and (**f**) a slightly reduced full-width-at-half-maximum (FWHM) energy spread in the accelerated spectrum (blue area) compared to the initial spectrum (orange area). All emittance measurements were performed using the high-resolution spectrometer.

dipole magnets; one for broad-band spectrum measurements on a gadolinium-oxysulfide (GadOx) screen situated outside the vacuum, and another for high-resolution, energy-resolved emittance measurements on an in-vacuum cerium-doped gadolinium-aluminium-gallium-garnet (GAGG:Ce) screen. Five quadrupoles were used to capture a point-to-point image of the electron beam from the plasma-cell exit (the object plane) to one of the two screens (the image plane).

## Characterization of the operating point

A multi-parameter optimization varying the incoming electron beam and the plasma density, as developed in a previous publication[14], resulted in the operating point visualized by the particle-in-cell (PIC) simulation shown in Fig. 1b (see "Methods" and Supplementary Fig. 10), which indicates a peak accelerating field of approximately 1.5 GV m$^{-1}$. The trailing bunch gained up to 40 MeV of energy per particle at an energy-transfer efficiency of around $22 \pm 2\%$ (see "Methods" and Supplementary Fig. 6), measured with the broad-band spectrometer (see Fig. 1c), and had a ~ 40 pC of charge both before and after acceleration, measured with the high-resolution spectrometer (see Fig. 1d, e). The reduced energy spread of the accelerated spectrum (see Fig. 1f), together with the observed high energy-transfer efficiency, indicate that the wakefield was strongly beam-loaded[35]. This effect is also observed in the PIC simulation, which indicates that the wakefield was under-loaded in the low-density ramp regions and overloaded in the high-density central region, resulting in an approximately uniform acceleration when longitudinally averaged (see "Methods"). Since a small low-energy distribution tail was introduced during acceleration, the energy spread is quantified using the full-width at half maximum (FHWM), as this correlates better with peak spectral density (the quantity most relevant to applications) compared to the more conventional rms (see "Methods").

## Preservation of emittance

Figure 2 demonstrates preservation of the projected (i.e., averaged over all energy slices), normalized emittance in the horizontal plane; starting at $2.85 \pm 0.07$ mm mrad, measured with the plasma cell extracted, and ending up at $2.80 \pm 0.09$ mm mrad after acceleration in the plasma. The root-mean-square (rms) horizontal beam size was measured across a range of object planes by varying the strength of the imaging quadrupoles (see "Methods" and Supplementary Fig. 7), while keeping a constant magnification as well as a constant object plane in

the vertical (dispersive) plane to ensure high-energy resolution (see Fig. 2b, c). This multi-shot measurement was only possible due to the high stability of the beam–plasma interaction (see Fig. 1c). The divergence was measured to be 0.28 mrad rms both before and after acceleration, with corresponding virtual-waist beam sizes of 5.0 and 4.7 μm rms. The screen resolution, measured to be 6.2 μm rms (see "Methods" and Supplementary Fig. 5), affected the measurement minimally, as the quadrupole imaging magnified the beam size by a factor of 7.9, thereby allowing sub-μm beam features to be resolved. The preservation of emittance was achieved simultaneously with that of charge and relative energy spread: these were within the 68th percentile range of their initial values in 41% and 62% of all shots, respectively (see Fig. 2d, e).

## Comparison to particle-in-cell simulations

The evolution of the beam inside the plasma was estimated using simulation (see Fig. 2a and Supplementary Fig. 10). This suggests that the trailing bunch was focused down to a beam size of less than 2 μm rms, undergoing 870° of phase advance (i.e., nearly five betatron envelope oscillations). The emittance was preserved even in the presence of a small mismatch of the beta function; the expected emittance growth after full phase mixing[25] is ~ 10%, but this was never reached because the decoherence length for a per-mille-level energy spread would be tens of metres. Moreover, since the driver was focused $21.3 \pm 0.3$ mm upstream compared to the trailing bunch (due to the chromaticity of the final-focusing quadrupoles) and had a higher emittance, the transverse size of the driver was relatively large, which both suppressed the hose instability[36] and resulted in negligible motion of argon ions on the timescale of one plasma oscillation. Emittance growth from Coulomb scattering, estimated analytically from the simulation to be $1.1 \times 10^{-4}$ mm mrad, was also negligible due to the small beta function inside the plasma cell[30].

## Emittance growth from misalignment

The main experimental challenge was to align and match the incoming bunch to the plasma wake. Misalignment and mismatching must be sufficiently small to avoid sampling the nonlinear focusing fields in the electron sheath surrounding the plasma cavity. The emittance-preserving operating point shown in Fig. 2 was found using high-precision scans of two key parameters: the angle between the trajectories of the driver and the trailing bunch (see Fig. 3), and the

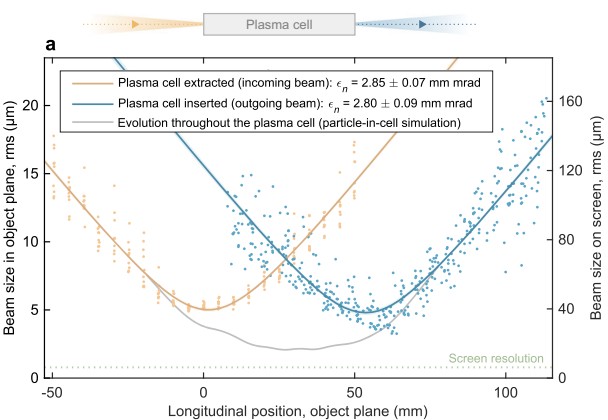

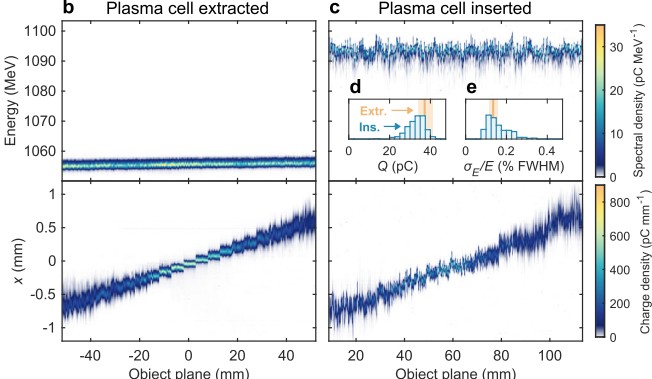

**Fig. 2 | Preservation of projected, normalized emittance. a** The imaged beam size is shown for a range of object planes around the plasma cell, measured with the plasma cell extracted (orange points) and inserted into the beam path (blue points). The screen resolution (green dotted line) is negligible. Note that the imaged beam size does not represent the beam size as it was inside the plasma cell, but instead that of the resulting virtual waist. Fits of the virtual-waist evolution (orange and blue lines) demonstrate that the normalized emittance, $\epsilon_n$, was preserved to within the

fit error. The evolution of the beam size throughout the plasma cell is estimated using a PIC simulation (grey line). **b, c** The measurement was performed by scanning the object plane of a point-to-point imaging spectrometer, first with the plasma cell extracted (**b**, 210 shots) and subsequently inserted (**c**, 420 shots); projections in energy and transverse position are displayed in the upper and lower panels, respectively. **d, e** The insets show the charge, $Q$, and relative energy spread, $\sigma_E/E$, before (orange lines) and after acceleration (blue histograms).

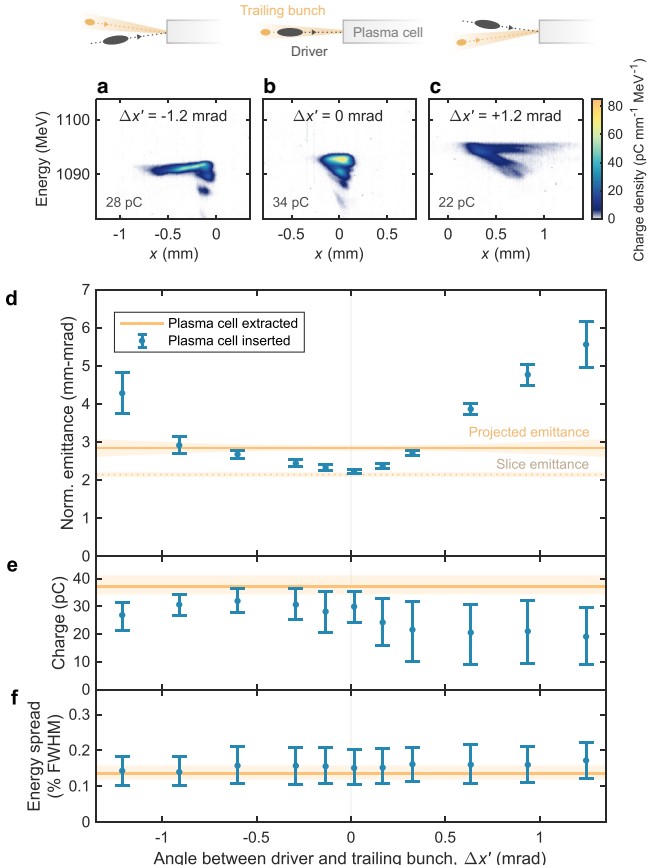

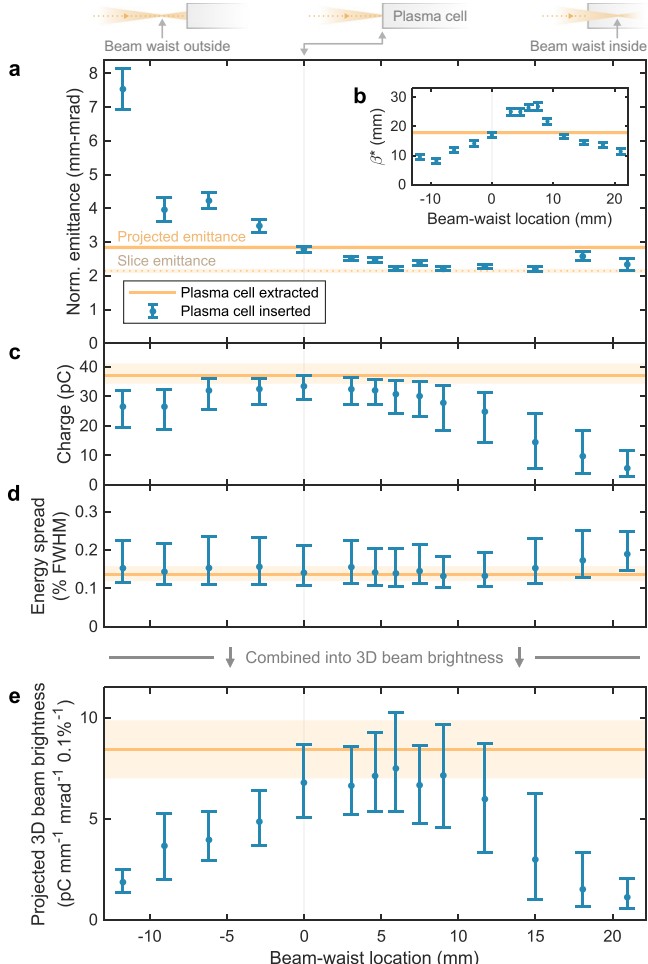

**Fig. 3 | Emittance growth due to misalignment. a–c** Spectrometer images, captured with the plasma cell inserted and at an object plane 65 mm upstream of the plasma-cell exit, show how the accelerated trailing bunch is distorted by misalignment (**a, c**) compared to optimal alignment (**b**). This scan was performed at a beam-waist location 2–7 mm downstream of that of the emittance-preserving operating point (see Fig. 2), resulting in a small charge loss around the optimal alignment. **d** Projected emittance measurements (blue error bars) are shown for a scan of angles between the driver and trailing bunches. The initial projected emittance (orange line), measured with the plasma cell extracted and at optimal alignment, where the error (light orange area) increases with misalignment to account for emittance growth from dispersion. The initial slice emittance is also shown (orange dotted line). **e** A somewhat asymmetric charge loss is observed. **f** The energy spread remained preserved throughout the scan. In (**d**), error bars represent the best-fit value and fit error, whereas in (**e, f**), they represent the median and 68th percentile range of the shot distributions.

**Fig. 4 | Evolution of beam qualities and 3D brightness with mismatching. a** The projected emittance (blue error bar) increased when the beam was focused upstream of the plasma-cell entrance. Focused downstream, the initial emittance (orange line) was preserved and even reduced down to the initial slice emittance (orange dotted line). **b** Throughout the scan, the virtual-waist beta function $\beta^*$ varied significantly, consistent with a change in matching. **c** With the beam waist at the plasma-cell entrance, the charge was preserved for roughly half the shots (see Fig. 2d), whereas as the waist moves away from the entrance, the charge is progressively lost. **d** The energy spread was similarly preserved for most scan steps. **e** Combining the above beam qualities, the projected 3D beam brightness was preserved for beam-waist locations in the range from 0 to +9 mm. In (**a, b**), error bars represent the best-fit value and fit error, whereas in (**c–e**), they represent the median and 68th percentile range of the shot distributions.

longitudinal waist location of the focused trailing bunch (see Fig. 4). At each point in these scans, an object-plane scan such as that shown in Fig. 2 was performed. This investigation required a fine-tuning of the quadrupoles used for alignment and matching at the 0.1% level: ~0.1 mrad in alignment and ~2 mm in waist location, respectively.

Figure 3 shows the effect of misalignment on the emittance. The angle between the driver and the trailing bunch was scanned by varying the horizontal dispersion with a quadrupole magnet in the upstream dispersive section. Since the mean energy of the two bunches was slightly different (by 0.9%), this dispersion resulted in a relative misalignment (by up to $\Delta x' = \pm 1.2$ mrad). However, because the corresponding range of quadrupole strengths ($\pm 1.5\%$) as well as the beam size in this quadrupole were both small, the beam-waist location remained within a range of 2–7 mm downstream of the location of the emittance-preserving operating point, while the waist beta function changed by less than $\pm 25\%$ (see "Methods" and Supplementary Fig. 2). The initial emittance was measured at optimal alignment ($\Delta x' \approx 0$). Modelling of the induced intra-bunch dispersion added up

to $\pm 6\%$ to the uncertainty of the projected emittance for misaligned bunches (see "Methods").

After inserting the plasma cell, optimal alignment resulted in a projected emittance somewhat lower than the initial projected emittance, close to the initial slice emittance of the central energy slice. This may be explained by an intrinsic intra-bunch dispersion within the trailing bunch (measured to be ~0.1 mrad per 0.1% of energy) that exists even when the driver and trailing bunch centroids were aligned. If the charge is lost from the tail of the trailing bunch during acceleration, which is consistent with observations (central error bar in Fig. 3e), this can reduce the intra-bunch dispersion and hence decrease the projected emittance. Note that this reduced emittance and charge does not contradict the preservation in Fig. 2, as the central point in the alignment scan had a beam-waist location 4.5 mm downstream of the emittance-preserving operating point. Further, the emittance was observed to grow with increased misalignment. The spectrometer

images in Fig. 3a and c, corresponding to large misalignments, show evidence of interaction with nonuniform focusing fields, deflecting the bunch tail and resulting in a higher charge loss. An asymmetry is also observed in Fig. 3d–qualitatively consistent with a simulated alignment scan (see Supplementary Fig. 13a), which indicates that it is caused by the small shift in waist location across the scan.

### Emittance growth from mismatching

Using optimally aligned bunches, the matching of the beam was varied. Figure 4 shows a scan of the beam-waist location across a 33 mm range around the plasma-cell entrance, performed by fine-tuning the strength of a final-focusing quadrupole (by ± 0.65%). The optical setup in the final-focusing section was such that the beam size in this quadrupole was much larger in the horizontal plane than in the vertical plane, allowing the horizontal and vertical waist locations to be adjusted independently. The driver- and trailing-bunch waist locations were measured separately at each step of the scan using a two-BPM measurement technique[37] where the distribution of the orbit jitter serves as a proxy for the beam (see "Methods" and Supplementary Fig. 2). This measurement also indicated that the relative separation between the waist locations of the driver (focused upstream) and the trailing bunch (focused downstream) remained fixed at 21.3 mm and that their waist beta functions stayed approximately constant throughout the scan.

The emittance was observed to increase dramatically when the beam was focused upstream of the plasma-cell entrance (see Fig. 4a). Conversely, when focused downstream, the emittance stayed approximately constant. However, in this case, significant loss of charge was observed for beam-waist locations beyond +10 mm (see Fig. 4c). This asymmetric behaviour may be explained by the accompanying change in driver focusing: focused upstream, the lower-density driver takes longer to establish a blowout cavity, which initially causes the trailing bunch to experience nonlinear focusing; focused downstream, the emittance-preserving blowout cavity is established immediately, but the large size of the mismatched trailing bunch causes it to lose charge from defocusing in the cavity walls. A simulation of the waist location scan (see Supplementary Fig. 13b) qualitatively reproduces the emittance growth in Fig. 4a.

### Preservation of beam brightness

A more unified comparison of all beam qualities before and after acceleration can be made using the projected three-dimensional (3D) beam brightness, calculated by dividing the peak spectral density by the projected emittance (see "Methods"). The simultaneous preservation of emittance, charge, and energy spread implies that the 3D beam brightness in the horizontal plane was preserved (see Fig. 4e); 39% of shots in a 9 mm range of waist locations fell within the 68th percentile range of the initial brightness. Moreover, bunch lengthening does not typically occur within a plasma accelerator, implying that the 4D brightness was also likely to have been preserved. Lastly, although it could not be measured in this experiment, the axial symmetry of a plasma accelerator suggests that emittance preservation can also be demonstrated in the vertical plane (see "Methods" and Supplementary Fig. 10), ultimately resulting in full 6D beam-brightness preservation.

## Discussion

While the emittance preservation achieved in this experiment was associated with modest energy gain, the techniques and achieved level of precision in alignment and matching (see Figs. 3 and 4) are consistent with those required for emittance preservation in a high-energy-gain plasma accelerator.

This conclusion is supported by simulation: starting from a PIC simulation that agrees with the experimental result (see Supplementary Fig. 10), identical input beams and plasma-density ramps were simulated but with the central flat-top density extended by 500 mm

(see "Methods")–emulating an FEL energy booster. The simulation shows significantly more energy gain (760 MeV) while still preserving the emittance to within the measurement error (Supplementary Fig. 11). Since the current profile of the trailing bunch was optimized for a shorter, non-uniform plasma-density profile, the wakefield is overloaded in the extended simulation, resulting in increased energy-transfer efficiency (33%) but also increased energy spread (1.5% rms). To show that this can be mitigated, another simulation was performed, using an identical driver and extended plasma, but shaping the trailing-bunch current profile to preserve the energy spread and to give a similar efficiency as in the experiment (see Supplementary Fig. 12). Simulated alignment and waist location scans, using both the optimized and non-optimized trailing bunches (see Supplementary Fig. 13), show that emittance is also preserved for high-energy gain, provided the bunches are aligned to within ± 0.1 mrad and the waist location placed within ± 5 mm of the optimum–consistent with the precision and sensitivity obtained in the current experiment. No transverse instabilities[38–40] were observed in this experiment, nor in the corresponding simulations (see Supplementary Fig. 10c); however, the normalized centroid offset[41] grew by ~ 47% in the simulation of the extended 500 mm plasma cell with optimal beam loading–sufficiently small not to affect the emittance in such a cell.

However, considering now very high energy gains and high energy efficiency[42], for example, as required for a plasma-based linear collider[43], such transverse instabilities may indeed be a major experimental challenge. Transverse instability of the driver (i.e., the hose instability[27]) can be mitigated by increasing its transverse beam size[36], as in this experiment, and further suppressed by the large energy spread induced by its deceleration[38]. These mitigation strategies cannot be applied to the trailing bunch, which must remain matched and maintain a low energy spread. However, the trailing-bunch instability can nevertheless be mitigated by detuning the betatron-oscillation frequency of different slices within the bunch. Methods include introducing an energy chirp[44] or a controlled amount of ion motion to provide nonuniform focusing within the bunch[45,46]–strategies that must be carefully balanced against their potential for additional emittance growth. Finally, applications such as a linear collider using electron drivers will require acceleration across multiple stages[47], which comes with a different set of challenges[48,49] and proposed solutions[50].

In summary, we have demonstrated that beam quality, and in particular, the emittance, energy spread and charge, can be simultaneously preserved in a plasma-accelerator stage. This is an essential milestone toward achieving compact, high-energy particle accelerators for applications, such as high-brightness FELs or high-luminosity linear colliders, where the performance critically depends on emittance and other beam qualities.

## Methods

### Electron driver- and trailing-bunch generation

The FLASH linac provided electron bunches with 880 pC of charge from a photocathode source, accelerated to 1050 MeV by superconducting RF cavities. The bunches were compressed with two magnetic chicanes to a bunch length of 285 μm rms, and approximately linearized in longitudinal phase space with a third-harmonic cavity. Active feedback for charge, energy, bunch length, and orbit was used to stabilize the operation over the multi-hour data-acquisition period. A double-bunch temporal structure was created by dispersing the electrons in energy in the horizontal plane onto two block collimators[34] that removed the high- and low-energy tails, as well as a notch collimator that split the bunch into a leading driver (400 pC) and a trailing bunch (40 pC).

### Transverse alignment and final focusing

Horizontal alignment of the two bunches was accomplished by adjusting a quadrupole and a sextupole, located downstream of the

collimators in a region of large horizontal dispersion, in order to cancel first- and second-order tilts[51]. Vertical alignment was not critical, as negligible vertical dispersion was introduced throughout the beam line. Final focusing was performed using nine quadrupoles in a 13 m-long straight section just downstream of the dispersive section. These quadrupoles were optimized to focus the beam on small beta functions while minimizing the first-order chromaticity in both planes (Supplementary Fig. 1). The longitudinal position of the beam waist, located close to the plasma-cell entrance, was precisely adjusted in the horizontal plane using the third-last quadrupole before the plasma cell, where the horizontal beta function was 6.8 times larger than in the vertical plane, and adjusted in the vertical plane using the penultimate quadrupole, where the vertical beta function was 7.0 times larger than in the horizontal plane. While the beam was strongly focused and the beam current was high, space-charge effects were nevertheless negligible due to the GeV-level particle energy.

## Beam-waist measurements

The location and beta function of the beam waist, as well as the relative misalignment between the driver and the trailing bunch, were estimated using a BPM-based measurement technique developed in a previous publication[37], where the multi-shot distribution of the orbit jitter is used as a proxy for the beam. Supplementary Fig. 2 shows these beam-waist parameters measured at each step of the scans shown in Fig. 3 (horizontal alignment scan) and Fig. 4 (horizontal beam-waist location scan). The fit in Supplementary Fig. 2c was used for angular calibration in Fig. 3, and the fit in Supplementary Fig. 2d was used for the calibration of beam-waist locations in Fig. 4. These measurements also show that the overall beam angle jitters by 0.1 mrad rms in the horizontal plane and 0.03 mrad rms in the vertical plane. Since the driver and trailing bunches are created from the same initial bunch, their angular jitter is expected to be correlated, resulting in a smaller jitter in the relative misalignment between the bunches.

## Longitudinal-phase-space measurements

The charge distribution of the electron bunches in longitudinal phase space was characterized using a PolariX-type[52] X-band RF transverse-deflection structure (TDS) placed 33 m downstream of the plasma cell. During this measurement, no beam–plasma interaction took place. The electron bunches were streaked vertically by the TDS and horizontally dispersed by a dipole magnet onto an in-vacuum GAGG:Ce screen. In order to maximize the resolution, three quadrupole magnets were used to point-to-point image in the dispersive plane, from the TDS to the measurement screen, and parallel-to-point image in the streaking plane. Supplementary Fig. 3 shows the longitudinal phase space of the double-bunch structure using a two-point tomographic reconstruction[53] based on both zero crossings in order to remove distortions caused by dispersion. This reconstruction shows that the driver had an average peak current of 1.0 kA and a bunch length of 42 μm rms (140 fs rms), while the trailing bunch had an average peak current of 0.44 kA and a bunch length of 11 μm rms (37 fs rms). The centroids of the two bunches were separated by 195 μm (650 fs). Individual shots (Supplementary Fig. 3a, b) show evidence of a microbunching instability[54]; however, these microbunches do not significantly affect the plasma wake as their wavelength of ~10 μm (~33 fs) is much shorter than the minimum plasma wavelength of ~300 μm (~1 ps).

## Plasma generation and density measurements

The plasma was generated using a discharge capillary[55], consisting of a 1.5 mm-diameter, 50 mm-long channel milled from two sapphire blocks. Argon gas doped with 3% hydrogen continuously flowed into the capillary via two gas inlets, located 2.5 mm from the entrance and exit. Using a backing pressure of 30 mbar, the resulting capillary pressure was 9 mbar, as measured with a pressure sensor connected close to the gas inlet. Short (400 ns flat-top), high-voltage (25 kV), high-current (400 A) discharge pulses between two electrodes at the capillary entrance and exit were used to ignite the plasma, after which the density decayed exponentially with a half-life of 2.1 μs (Supplementary Fig. 4a). This was measured using spectral-line broadening of the H-alpha line[56], observed with an optical spectrometer collecting light from the full capillary radius in a 7 mm longitudinal region near the centre of the plasma cell, integrating over 0.2 μs on an intensified camera. The beam arrived 9.68 μs after the initial discharge, at which time the radially averaged plasma density at the centre of the plasma cell had decayed to approximately $8 \times 10^{15}\,\mathrm{m^{-3}}$. The longitudinal plasma-density profile, measured by displacing the cell longitudinally[57], is consistent with a Gaussian-like density profile (Supplementary Fig. 4b). The radial plasma-density profile was not measured, but the on-axis density during the period of exponential decay (beyond ~2 μs) is expected to be approximately 50% higher than the measured average (i.e., $1.2 \times 10^{16}\,\mathrm{m^{-3}}$); this effect is observed when electron beams are translated radially inside the capillary, as well as in magnetohydrodynamic simulations[58]. Low-density longitudinal ramps outside the plasma cell, which can affect the beta function, could also not be measured. Nevertheless, we assume an inverse-square profile (reaching half density 4 mm outside the electrodes of the cell) based on observed cone-shaped light emissions.

## Broadband and high-resolution imaging spectrometers

Two electron-energy spectrometers were used in this experiment, one for broad-band spectrum measurements and another for the high-resolution emittance measurements, both using a 1.07 m-long vertically dispersive dipole magnet. Five quadrupole magnets with a 5 mm bore radius were used to point-to-point image the diverging electron bunches from the plasma cell to the measurement screens. The broad-band spectrometer used an out-of-vacuum scintillator screen (GadOx), giving a spatial resolution of approximately 50 μm rms, placed 4 m downstream of the plasma cell, resulting in a horizontal beam-imaging magnification of a factor −3. The high-resolution spectrometer used an in-vacuum scintillator screen (GAGG:Ce) located 7.3 m downstream of the plasma cell, resulting in a larger horizontal magnification of −7.9. The corresponding vertical magnification was approximately −2.6 (for ~1.05 GeV) and −2.7 (for ~1.1 GeV). The resolution of this screen (part of a European XFEL-type screen station[59]), imaged with Scheimpflug optics, was measured to be 6.2 μm rms or smaller (Supplementary Fig. 5) by imaging a beam focused to less than 5 μm rms using an imaging optic with a magnification of −1. The pixel size, corresponding to $5.5 \times 5.5\,\mathrm{\mu m^2}$ on the screen, does not contribute significantly to the resolution but was nevertheless accounted for as part of the above resolution measurement. A charge-density calibration was performed on both screens by scanning the position of an energy collimator and correlating the integrated on-screen charge with that measured in an upstream toroidal current transformer. Scintillator saturation effects in the high-resolution screen were accounted for by correcting the light yield by Birk's law[60]

$$\rho = \frac{\rho_{\mathrm{scint}}}{1 - B\rho_{\mathrm{scint}}}, \qquad (1)$$

where $\rho$ is the true charge density, $\rho_{\mathrm{scint}}$ is the apparent charge density measured on the scintillator, and $B$ is Birk's constant. The material GAGG:Ce was chosen specifically for its high saturation threshold[61], which was estimated experimentally to be approximately 18 nC m$^{-2}$, or equivalently to a $B = 0.056\,\mathrm{m^2\,nC^{-1}}$. However, since the peak charge density obtained during emittance measurements (i.e., while point-to-point imaging the virtual waist) was never above 2.7 nC m$^{-2}$, this correction is small and has only a percent-level effect on the average measured charge.

## Energy-transfer-efficiency measurements

The energy-transfer efficiency was calculated to be 15–25% (with a distribution mode of 22%) by comparing the energy gained by the trailing bunch to the energy lost by the driver,

$$\eta = -\frac{\Delta E_{\text{trailing}} Q_{\text{trailing}}}{\Delta E_{\text{driver}} \bar{Q}_{\text{driver}}}, \tag{2}$$

where $\Delta E$ denotes the change in mean energy, and $Q$ is the charge: the final charge of the trailing bunch, and the mean of the measured initial and final charge for the driver (i.e., the best estimate in case of missing driver charge after deceleration). The distribution of efficiency is shown in Supplementary Fig. 6. For improved accuracy, the average driver spectrum was reconstructed from a scan of imaging energies, making use only of the part of the spectrum closest to the correctly imaged energy.

## Emittance measurements

The emittance of the trailing bunch was measured by scanning the strength of the imaging quadrupoles such that only the horizontal object plane changed, whereas the horizontal magnification and vertical object plane (for good energy resolution) remained constant, as shown in Supplementary Fig. 7. A slight vertical offset of one or more of these quadrupoles led to a slight shift in the apparent energy throughout these object-plane scans (as seen in the energy projection in Fig. 2b). The object plane and magnification were calculated for each shot individually based on the measured mean energy of the trailing bunch and the current in each quadrupole. The true beam size (i.e., at the location of the object plane) was calculated by dividing the measured beam size on the screen by the magnification. The screen resolution had a negligible effect, increasing the measured beam size by 1.5% or less. The normalized emittance $\epsilon_n$, waist beta function $\beta^*$, and beam-waist location $s^*$ were extracted by fitting a ballistic beam-waist envelope model to the measured true beam size

$$\sigma(s) = \sqrt{\frac{\epsilon_n}{\gamma}\left(\beta^* + \frac{(s - s^*)^2}{\beta^*}\right) + \left(\frac{\sigma_{\text{res}}}{M}\right)^2}, \tag{3}$$

where $s$ denotes the object plane, $\sigma_{\text{res}}$ denotes the screen resolution, and $M$ denotes the beam-imaging magnification. The exact magnification was $M = -7.87$ (with 0.03% rms jitter) with the plasma cell extracted (see Fig. 2b), and $M = -7.88$ (with 0.15% rms jitter) with the plasma cell inserted (see Fig. 2c), where the increased jitter is caused by the energy jitter resulting from the plasma acceleration. For the emittance measurements used for Figs. 3 and 4 (shown in full in Supplementary Figs. 8 and 9), the incoming bunch length and the beam charge (combined driver and trailing bunch charge) were filtered to only include a range ± 15% and ± 5%, respectively, to ensure similar input parameters throughout the multi-hour measurement. Here, the bunch length was measured prior to double-bunch generation (i.e., notch collimation) with a calibrated pyroelectric detector for coherent diffraction radiation. Lastly, in addition to the measurement uncertainty, the incoming projected emittance shown in Fig. 3 has an uncertainty related to dispersion induced by misalignment: $D_{x'} \approx \Delta x' / (\delta E / E)$, where $\Delta x'$ and $\delta E / E = 0.9\%$ are the relative angle and relative energy difference between the driver and trailing bunches, respectively. This dispersion, when multiplied by the relative energy spread $\sigma_\delta \approx 0.06\%$ rms of the trailing bunch, adds/subtracts in quadrature with the measured divergence $\sigma_{x'} = 0.28$ mrad rms, resulting in a relative emittance uncertainty

$$\frac{\sigma_\epsilon}{\epsilon} = \sqrt{1 + \left(\frac{\sigma_\delta D_{x'}}{\sigma_{x'}}\right)^2} - 1, \tag{4}$$

corresponding to a 6% added uncertainty at maximal misalignment ($\Delta x' = \pm 1.2$ mrad).

## Projected 3D-beam-brightness calculations

The projected 3D beam brightness, as shown in Fig. 4e, is calculated using the formula

$$B_{\text{3D}} \equiv \frac{1}{\epsilon_{nx}}\left(\frac{\partial Q}{\partial \delta}\right)_{\text{peak}}, \tag{5}$$

where the peak of the relative spectral charge density, $\partial Q / \partial \delta$, is divided by the projected normalized emittance in the horizontal plane, $\epsilon_{nx}$. Here, $\delta = \Delta E / E$ is defined as the relative energy offset. The uncertainty of the 3D brightness, shown as error bars in Fig. 4e, is estimated by Monte-Carlo sampling: dividing the peak spectral density of all the shots in each step by a large number of normally distributed samples of the emittance in that step (whose distribution is defined by the best-fit value and error), and then quantifying the width of the resulting 3D brightness distribution as the 68th percentile range (equivalent to ± 1 sigma if the distribution would be normal, which it is not).

## Particle-in-cell simulations

Particle-in-cell simulations were performed using the open-source 3D code HiPACE++[62], which uses the quasi-static approximation. The input beam was reconstructed in 6D phase space based on beam-waist measurements using BPMs (Supplementary Fig. 2), longitudinal-phase-space measurement using a TDS (Supplementary Fig. 3), as well as the measured transverse phase space of the trailing bunch (see Fig. 2). The horizontal and vertical slice emittances of the driver were not measured, but kept as free parameters. The longitudinal plasma-density profile was based on the optical spectrometer measurement, with assumed external ramps (Supplementary Fig. 4). Since the incoming vertical emittance of the trailing bunch could not be measured on the spectrometer, it was assumed to be identical to that of the incoming horizontal emittance—roughly consistent with previous measurements elsewhere in the linac. Simulations were performed in a box of size $388 \times 388 \times 379$ µm³ in the horizontal, vertical, and longitudinal directions, respectively, with 0.38 µm resolution (i.e., $1023 \times 1023 \times 998$ grid cells). The step size was 110 µm (366 fs). The beam was resolved with 4 million constant-weight macroparticles; the plasma was initialized with zero temperature and resolved with 1 particle per cell. The trailing bunch was horizontally misaligned by 0.05 mrad about the entrance of the plasma cell. The simulation results (Supplementary Fig. 10) are consistent with all the experimental measurements: the charge and the projected normalized emittance in the horizontal plane are both preserved, while the energy spread was slightly reduced. The simulation also suggests that the emittance was (or could in principle be) preserved also in the vertical plane.

## Simulations with increased energy gain

In order to assess the scalability of the measured emittance preservation, a PIC simulation with an extended plasma cell was performed. Here, both the input beam parameters and the plasma-density profile in the up and down ramps were identical to that used in the shorter simulation (Supplementary Fig. 10). However, a central 500 mm flat-top density region was introduced (between the up and down ramps) to increase the trailing-bunch energy gain to near energy doubling: 758 MeV per particle. The same resolution, step size and number of beam particles as in the shorter simulation was used. The results are shown in Supplementary Fig. 11. Here, the emittance in the horizontal plane is observed to increase by only 1.6%—preserved to within the measurement error (± 3%). Similarly, in the vertical plane, the simulation indicates an emittance growth of 1.3%. While the trailing bunch undergoes 8000° of phase advance (i.e., more than 44 betatron envelope oscillations), no transverse instabilities are observed to cause

emittance growth. The argon ions were mobile, but their motion was negligible. Coulomb scattering was not included in this simulation but can be estimated analytically to increase the emittance by $\sim 1.2 \times 10^{-3}$ mm mrad, which is negligible. In this simulation, the peak spectral density decreases significantly, as the trailing-bunch current profile is not optimized for the plasma-density profile of the extended cell, but instead to flatten the longitudinally averaged wakefield across the shorter cell. Consequently, the trailing bunch overloads the wakefield in the extended flat-top region, leading to higher energy-transfer efficiency (33%) but also a chirped distribution in longitudinal phase space with a 1.5% rms energy spread. To preserve the incoming energy spread and to give a similar efficiency, another simulation was performed using an identical driver and extended plasma-density profile, but an optimized current profile, shown in Supplementary Fig. 12. In this simulation, the emittance, charge and energy spread were all preserved.

### Simulated scans of misalignment and mismatching

Simulations of the misalignment scan (as shown in Fig. 3) and beam-waist location scan (as shown in Fig. 4) were performed, as shown in Supplementary Fig. 13. For reduced computational load, these simulations were performed at half resolution (i.e., 0.76 µm across $511 \times 511 \times 499$ grid cells), but were otherwise identical. The misalignment scan consists of 41 simulations from $-1.2$ to 1.2 mrad (0.06 mrad per step), qualitatively recreating the shape of the emittance growth measured in the experiment. For the extended plasma cells, both with and without the optimized current profile, the final emittance is larger but remains preserved in a $\pm 0.07$ mrad range around zero. To properly resolve this central region, a fine scan was performed between $-0.2$ and 0.2 mrad ($\sim 0.01$ mrad per step). Similarly, the beam-waist location scan consists of 41 simulations ranging from $-20$ mm upstream to 20 mm downstream of the plasma-cell entrance (2 mm per step). Again, the experiment is qualitatively recreated. In the extended plasma cell, the emittance growth is of similar order and sensitivity as in the shorter cell; the emittance is preserved within a $\pm 5$ mm range of the optimum.

## Data availability

The experimental data generated in this study have been deposited at https://doi.org/10.5281/zenodo.11967839.

## Code availability

HiPACE++ (v23.11) is openly available at https://github.com/Hi-PACE/hipace and also at https://doi.org/10.5281/zenodo.5639467. All analysis and simulation scripts have been deposited at https://doi.org/10.5281/zenodo.11967839.

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

## Acknowledgements

We thank M. Dinter, S. Karstensen, S. Kottler, K. Ludwig, F. Marutzky, A. Rahali, V. Rybnikov, A. Schleiermacher, the FLASH management, and the DESY FH, M and FS divisions for their scientific, engineering and technical support. This work was supported by Helmholtz ARD, Helmholtz ATHENA, the Helmholtz IuVF ZT-0009 programme and the Maxwell computational resources at DESY. C.A. was funded by the Research Council of Norway (NFR Grant No. 313770) and the European Research Council (project SPARTA, ERC Grant Agreement No. 101116161). We acknowledge Sigma2 - the National Infrastructure for High-Performance Computing and Data Storage in Norway for awarding this project access to the LUMI supercomputer, owned by the EuroHPC Joint Undertaking, hosted by CSC (Finland) and the LUMI consortium. The authors also gratefully acknowledge the Gauss Centre for Supercomputing e.V. (www.gauss-centre.eu) for funding this project by providing computing time through the John von Neumann Institute for Computing (NIC) on the GCS Supercomputer JUWELS at Jülich Supercomputing Centre (JSC).

## Author contributions

C.A.L., R.D., and J.O. conceived the experiment. C.A.L., L.B., J.B.S., F.P. and J.C.W. performed the experiment, with help from J.B., J.C., R.D., J.M.G., P.G.C., G.L., S.S., and S.W. J.M.G and G.L. performed the plasma-density measurements. C.A.L. analysed the experimental data and produced all the figures. C.A.L. wrote the manuscript, with assistance from B.F. C.A.L. performed the 6D beam reconstruction and corresponding HiPACE++ PIC simulations, with help from S.D. and M.T., who also developed new HiPACE++ features for these simulations. R.D. and J.O. supervised the project and the personnel. J.B. and J.C. were supervised by M.W. All authors discussed the results in the paper.

## Funding

## Competing interests

The authors declare no competing interests.
