## [Peer Review File · Nature Communications]

Emittance preservation in a plasma-wakefield acceleratorREVIEWER COMMENTS

Reviewer #1 (Remarks to the Author):

- What are the noteworthy results?

The authors measure the emittance of an electron beam with and without plasma acceleration and show that the measured emittance with and without acceleration is consistent with no change in emittance. They further measure the sensitivities of emittance preservation due to several common scenarios, variation in drive-witness input angles and drive-witness offsets. This manuscript documents a very well performed experiment, one of the best I've ever seen. With the caveats mentioned below, I recommend the paper for publication after revision.

- Does the work support the conclusions and claims, or is additional evidence needed?
- Are there any flaws in the data analysis, interpretation and conclusions? Do these prohibit publication or require revision?

In the abstract, the authors state "emittance preservation in a high-gradient, and high-efficiency PWFA". Whereas, in the introduction, the authors write "However, preservation of emittance at the level required for scaling to large energy gain has until now not been established." In the discussion section the authors write "modest energy gain". The measurement shown here is an energy increase of approximately 5%. I do not believe 5% is sufficient to claim "large energy gain". I do not find the extension to longer plasma through simulation compelling. I do not think that charge is preserved.

I do not agree with the statement in the first paragraph of the discussion "the plasma was sufficiently long to require the same techniques and level of precision in the alignment and matching that would also be require in longer plasma cells necessary for energy gain." On page 4 the authors describe that the charge loss as due to intra-bunch dispersion within the trailing bunch. The simulation shown in Supplementary Figure 11 doesn't discuss charge loss. Does the simulation include the intra-bunch dispersion? Such mismatches usually scale at least with the length of the accelerator, so I would expect the longer plasma to result in more charge loss.

- Is the methodology sound? Does the work meet the expected standards in your field?
The methodology is excellent. It easily meets the standard of the field.

- Is there enough detail provided in the methods for the work to be reproduced?
Yes, the paper is remarkably detailed.

- Will the work be of significance to the field and related fields? How does it compare to the established literature? If the work is not original, please provide relevant references.

Proponents will say it is a significant and necessary demonstration that proves the viability of plasma-based accelerators. Opponents will say that the overall energy gain of 5% is too small to address any of their concerns of the practicality of plasma-based accelerators. This is an excellent experiment and manuscript that provides data for all participants in the discussion. I recommend publication after revision.

Background:

Plasma-based accelerators are desirable because they "in principle make accelerators shorter and cheaper" [from manuscript]. A lot of ink has been spilled on the subject of emittance preservation in a plasma-based accelerator, in fact in all wakefield driven accelerators [manuscript ref 40]. The question that needs to be answered by the accelerator community is whether emittance can be preserved for significant luminosity at significantly reduced cost, not whether emittance can be preserved in any

scenario. This manuscript shows the latter.

Given the scaling in [manuscript ref 41] one would not expect significant emittance growth given the achieved gradient and admirable tolerances on the input beam parameters in this experiment. Additionally, the experiment does not reach levels of energy gain relevant for FELs or colliders, nor does it demonstrate emittance preservation using gradients typically proposed for compact accelerators for a collider application [<https://arxiv.org/abs/2303.10150>]. The scaling is as frequency^3 for transverse instabilities while acceleration scales as frequency^2 , so successful emittance at one gradient (tied to the frequency by the nature of a plasma-based accelerator) does not mean much for higher gradients. It would be instructive to reproduce plot 3d from the manuscript using the longer simulation.

Recommendation:

Publish after revision and re-review.

Specific comments and revision requests:

The paper should be adjusted to remove claims of large or significant energy gain.

The last sentence of the abstract should be removed. The second to last sentence should read "We demonstrate, for the first time, emittance preservation in a high-gradient and high-efficiency PWFA." The claim of charge preservation is questionable, and losses shown in Figure 2 would not be acceptable in energy boosters for photon science or for particle colliders. The remainder of the manuscript should also remove references to charge preservation.

A few small comments that can easily be corrected:

- 1) Reference 27 was published in 1991.
- 2) Page 7, bottom left, there is a line about "extended data fig. 8 and 9", is this supposed to be "supplemental"?

Reviewer #2 (Remarks to the Author):

The article by C.A. Lindström et al. presents a detailed study on emittance preservation in a plasma wakefield accelerator. It is well-written, clear, and didactic. The achieved control of the beam is impressive, but the main results seem to me too technical to be published in Nature Communications; they seem more suitable for a specialized audience.

Emittance preservation is one of the main challenges of PWFA accelerators, which has been thoroughly studied theoretically, showing a trade-off between efficiency and instability. Several techniques have been proposed to mitigate the emittance growth when it occurs. Here, emittance preservation is demonstrated in a small gain regime where no significant emittance growth is expected, for matched and well-aligned beams. The authors showed their ability to fulfill these conditions. They also measured tolerances for mismatching and misalignment, which is a significant result for specialists, but a general audience learns little: the article demonstrates the preservation of emittance in a case where there is no doubt that it should be preserved.

All the open questions on emittance preservation and the potential of beam-driven wakefield acceleration for building a collider remain open. This is highlighted in the discussion at the end of the article that evokes this trade-off between efficiency and instability, stresses the importance of the beam breakup instability, and methods for mitigating this phenomenon. Demonstrating one of these solutions would constitute a crucial step toward compact, high-energy particle accelerators for applications such as high-brightness FELs or high-luminosity linear colliders, but this article does not.

DESY | Hamburg

Dr. Carl A. Lindstrøm

(Former) Research Fellow, FTX-AST, FH, DESY
carl.a.lindstroem@desy.de

(Current) Researcher, Dept. of Physics, University of Oslo
c.a.lindstrom@fys.uio.no

10 May 2024

Response to the Reviewers

We would like to thank the reviewers for their work in evaluating our manuscript, as well as for their patience. In order to properly answer the reviewers questions, a large-scale simulation campaign was carried out, which involved the development of a new generation- and analysis framework, the performing of hundreds of full-scale 3D PIC simulations, and the use of in total several thousand GPU compute hours. We believe that the resulting additions have significantly improved the manuscript, in particular by exemplifying exactly how the experiment is relevant to future experiments with larger energy gain.

Quoted below are the reviewers' comments (in blue) and our response to each of them (in black). Changes to the manuscript are indicated in green.

Reviewer #1:

• What are the noteworthy results?

The authors measure the emittance of an electron beam with and without plasma acceleration and show that the measured emittance with and without acceleration is consistent with no change in emittance. They further measure the sensitivities of emittance preservation due to several common scenarios, variation in drive-witness input angles and drive-witness offsets. This manuscript documents a very well performed experiment, one of the best I've ever seen. With the caveats mentioned below, I recommend the paper for publication after revision.

We thank the reviewer for this positive evaluation of our work.

• Does the work support the conclusions and claims, or is additional evidence needed?

• Are there any flaws in the data analysis, interpretation and conclusions? Do these prohibit publication or require revision?

In the abstract, the authors state "emittance preservation in a high-gradient, and high-efficiency PWFA". Whereas, in the introduction, the authors write "However, preservation of emittance at the level required for scaling to large energy gain has until now not been established." In the discussion section the authors write "modest energy gain". The measurement shown here is an energy increase of approximately 5%. I do not believe 5% is sufficient to claim "large energy gain".

We fully agree that a 5% energy gain is not sufficient to be qualify as a "large" energy gain. This is why we instead call it "modest". The quoted sentence ("*However,*

Deutsches Elektronen-Synchrotron DESY
Notkestraße 85, 22607 Hamburg, Germany

Location Zeuthen
Platanenallee 6, 15738 Zeuthen, Germany

www.desy.de

Board of Directors

Prof. Dr. Helmut Dosch
(Chairman)

Christian Harringa
(Deputy Chairman)

Prof. Dr. Beate Heinemann

Prof. Dr. Wim Leemans

Prof. Dr. Christian Stegmann

Prof. Dr. Edgar Weckert

Dr. Arik Willner, CTO
(Delegate of the
Directorate for Innovation)

preservation of emittance at the level required for scaling to large energy gain has until now not been established") does not claim large energy gain. Instead, this paper discusses reaching the *level of precision* required to scale up plasma accelerators to be longer, while still preserving emittance. To avoid this confusion, we have altered the quoted sentence to simply say:

(Line 56): "However, preservation of emittance has until now not been established".

To clarify this point and to justify why the experiment was performed in a moderate-length plasma cell, we have added a new paragraph at the end of the Introduction:

(Line 90): "Not only is it challenging to preserve emittance, it is also non-trivial both to locate the ideal operating point and to measure accurately the emittance and energy spectrum. In practice, this difficulty scales with energy gain, because the ideal operating region shrinks and the larger (absolute) energy jitter of the accelerated bunch increases the difficulty of making accurate multi-shot emittance measurements. An initial demonstration of emittance preservation is therefore best carried out in a plasma accelerator that is long enough to display the relevant sources of emittance growth and be sensitive to the required tuning precision, but short enough to be compatible with current state-of-the-art stability in electron-beam and plasma generation."

I do not find the extension to longer plasma through simulation compelling. I do not think that charge is preserved.

I do not agree with the statement in the first paragraph of the discussion "the plasma was sufficiently long to require the same techniques and level of precision in the alignment and matching that would also be require in longer plasma cells necessary for energy gain." On page 4 the authors describe that the charge loss as due to intra-bunch dispersion within the trailing bunch. The simulation shown in Supplementary Figure 11 doesn't discuss charge loss. Does the simulation include the intra-bunch dispersion? Such mismatches usually scale at least with the length of the accelerator, so I would expect the longer plasma to result in more charge loss.

The evolution of the charge was not shown for any of the simulations, but the charge is in fact preserved in all the simulations, including those in the extended plasma cell. To show this, we have now included the charge evolution in Supplementary Figs. 10 and 11, as reproduced below in Figs. R1 and R2.

In the experiment, the charge was preserved for the emittance-preserving working point (Fig. 2)—not for every shot, but for 41% of the shots (as shown by the histogram in Fig. 2d). The charge was not preserved in every shot because there was a small jitter in both beam alignment and beam matching. Further, we realize there may be a misunderstanding in interpreting Fig. 3: this scan did not include the emittance-preserving working point shown in Fig. 2. While this fact was mentioned in the caption of Fig. 3 ("This scan was performed at a beam-waist location 2–7 mm downstream of that of the emittance-preserving operating point (see Fig. 2), resulting in a small

Fig. R1 (Supplementary Fig. 10): Updated version of the experiment-matched simulation figure, showing 100% charge preservation in subfigure (d).

Fig. R2 (Supplementary Fig. 11): Updated version of the extended-length simulation figure, showing 100% charge preservation in subfigure (d).

charge loss around the optimal alignment”) we have now addressed this point also in the manuscript text:

(Line 253): “Note that this reduced emittance and charge does not contradict the preservation in Fig. 2, as the central point in the alignment scan had a beam-waist location 4.5 mm downstream of the emittance-preserving operating point.”

The simulations do not directly include the intra-bunch dispersion (i.e., an $z-x'$ correlation), but instead include it indirectly by using the projected emittance as the slice emittance (with no $z-x'$ correlation). Given that we do not have a sufficiently accurate 6D reconstruction of the substructure within the trailing bunch (i.e., of the correlations between the centroid transverse offset, angle, energy and longitudinal position), it is not meaningful to include such correlations in the simulations—it would simply be tuning of free parameters to fit data. This would require very large simulation resources, but add little value.

Finally, the statement “mismatches usually scale at least with the length of the accelerator” is not completely correct. Assuming that a “mismatch” here refers to either a misalignment or a beta-function mismatch, it is not the case that the charge loss is expected to scale linearly with plasma length or energy gain. The amplitude of oscillation of each individual particle is either constant (for constant energy) or decreasing (for increasing energy, due to adiabatic damping). When operating within the linear fields of the plasma cavity, the sensitivity to beta-function mismatching is therefore instead expected to plateau after the first betatron oscillations. For large misalignment or mismatching, parts of the bunch will experience nonlinear focusing in the plasma-cavity walls, resulting in emittance growth and charge loss—an effect that is accentuated in longer plasmas (as the nonlinear fields are traversed for longer). Additionally, it is indeed true that even within the linear fields, a beam-breakup instability can cause the amplitude to grow, but this instability does not develop sufficiently on the length scale used in either the experiment or the extended simulation to cause any additional emittance growth or charge loss.

To illustrate the above point more clearly, and not only through physics-motivated arguments, we recreated the experimental scans performed in Figs. 3 and 4, both with the 50 mm plasma cell and for a cell extended by 500 mm. The result is shown in Figure R3 (Supplementary Fig. 13 in the manuscript). Firstly, we note that the scans qualitatively recreate the emittance growth observed in the experiment (blue curves). The extended simulations (green curves) show that a misalignment indeed produces a larger emittance growth, but that the emittance is preserved (to within the 3% measurement accuracy) in a range ± 0.1 mrad around optimal alignment, which matches both the tuning precision and the measurement sensitivity of the experiment. In the mismatching scans (varying the beam-waist location) a similar level of emittance growth is observed, indicating that the emittance can be preserved also for large energy gains as long as the beam waist is placed within ± 5 mm of the optimum—again consistent with the precision and sensitivity of the experiment.

Fig. R3 (Supplementary Fig. 13): Simulated misalignment and beam-waist location scans, matching those performed in the experiment, but also including the extended simulations.

We have changed the Discussion section to more clearly reflect the above discussion, and to emphasise the relevance of the experiment to higher-gain plasma accelerator applications:

(Line 319): “While the emittance preservation achieved in this experiment was associated with modest energy gain, the techniques and achieved level of precision in alignment and matching (see Figs. 3 and 4) are consistent with those required for emittance preservation in a high-energy-gain plasma accelerator.

This conclusion is supported by simulation: starting from a PIC simulation that agrees with the experimental result (see Supplementary Fig. 10), identical input beams and plasma-density ramps were simulated but with the central flat-top density extended by 500 mm (see “Methods”)—emulating an FEL energy booster. The simulation shows significantly more energy gain (760 MeV) while still preserving the emittance to within the measurement error (Supplementary Fig. 11). Since the current profile of the trailing bunch was optimized for a shorter, non-uniform plasma-density profile, the wakefield is overloaded in the extended simulation, resulting in increased energy-transfer efficiency (33%) but also increased energy spread (1.5% rms). To show that this can be mitigated, another simulation was performed, using an identical driver and extended plasma, but shaping the trailing-bunch current profile to preserve the energy spread and to give a similar efficiency as in the experiment (see Supplementary Fig. 12). Simulated alignment and waist-location scans, using both the optimized and non-optimized trailing bunches (see Supplementary Fig. 13), show that emittance is preserved also for high energy gain, provided the bunches are aligned to within ± 0.1 mrad and the waist location placed within ± 5 mm of the optimum—consistent with the precision and sensitivity obtained in the current experiment. No transverse instabilities^{38,39} were observed in this experiment, nor in the corresponding simulations (see Supplementary Fig. 10c); however, the normalized centroid offset⁴⁰ grew by $\sim 47\%$ in the simulation of the

extended 500-mm plasma cell with optimal beam loading—sufficiently small not to affect the emittance in such a cell.”

As discussed in this paragraph, an additional simulation was added to the Supplementary Material, showing acceleration in the extended cell of an optimally beam-loaded trailing bunch, resulting also in energy-spread preservation (see Fig. R3 below; Supplementary Fig. 12 in the manuscript).

• Is the methodology sound? Does the work meet the expected standards in your field?

The methodology is excellent. It easily meets the standard of the field.

• Is there enough detail provided in the methods for the work to be reproduced?

Yes, the paper is remarkably detailed.

We thank the reviewer for recognising the large amount of work that has gone into the methodology and detailed characterization of the beam and plasma. Ultimately, this reflects the key takeaway from the paper: while it was expected that emittance preservation could be achieved, doing so in practice requires an extremely precise understanding and control of the experimental setup.

Fig. R3 (Supplementary Fig. 12): Simulation of the extended cell, using an identical driver and plasma density profile, but with an optimally beam loaded trailing bunch. Here, in addition to the emittance, the energy spread is also preserved.

• Will the work be of significance to the field and related fields? How does it compare to the established literature? If the work is not original, please provide relevant references.

Proponents will say it is a significant and necessary demonstration that proves the viability of plasma-based accelerators. Opponents will say that the overall energy gain of 5% is too small to address any of their concerns of the practicality of plasma-based accelerators. This is an excellent experiment and manuscript that provides data for all participants in the discussion. I recommend publication after revision.

Background:

Plasma-based accelerators are desirable because they “in principle make accelerators shorter and cheaper” [from manuscript]. A lot of ink has been spilled on the subject of emittance preservation in a plasma-based accelerator, in fact in all wakefield driven accelerators [manuscript ref 40]. The question that needs to be answered by the accelerator community is whether emittance can be preserved for significant luminosity at significantly reduced cost, not whether emittance can be preserved in any scenario. This manuscript shows the latter.

Given the scaling in [manuscript ref 41] one would not expect significant emittance growth given the achieved gradient and admirable tolerances on the input beam parameters in this experiment. Additionally, the experiment does not reach levels of energy gain relevant for FELs or colliders, nor does it demonstrate emittance preservation using gradients typically proposed for compact accelerators for a collider application [<https://arxiv.org/abs/2303.10150>]. The scaling is as frequency³ for transverse instabilities while acceleration scales as frequency², so successful emittance at one gradient (tied to the frequency by the nature of a plasma-based accelerator) does not mean much for higher gradients. It would be instructive to reproduce plot 3d from the manuscript using the longer simulation.

As requested, and as discussed above, we have reproduced the misalignment scan for both the experimental and extended plasma cells (Figure R3 and Supplementary Fig. 13). These do not indicate any emittance growth due to transverse instability.

It is true that the demonstrated gradient (1.5 GV/m peak) was lower than that proposed for the HALHF concept [Foster *et al.*, New J. Phys. 25, 093037 (2023)] (by a factor ~5), but it was ~7 times higher than in the recent demonstration of free-electron lasing by a PWFA-boosted beam [Pompili *et al.*, Nature 605, 659 (2022)]. Moreover, further optimization of the HALHF parameter set (to be published) shows that the most cost-effective accelerating gradient in the plasma stages is approximately 2 GV/m—as the increased length of the plasma linac implied by this reduction in gradient is still negligible compared to the overall collider length at this gradient, but difficulties related to emittance growth and stability are much reduced. The plasma density in the original HALHF parameter set was $7 \times 10^{15} \text{ cm}^{-3}$ (lower than what was used in this experiment), but the cost optimization reduced this closer to $1\text{--}2 \times 10^{15} \text{ cm}^{-3}$. Lastly, while it is true that the gradient is tied to the frequency of the plasma-based accelerator, the argumentation employed by the reviewer is misleading in this case: it is also possible to increase the gradient by driving a more nonlinear wakefield (which is the case for HALHF), in which case the effective frequency is in fact *decreased* rather than increased (i.e., the wake becomes longer and its radius is larger). In sum, our view is therefore that the experiment was *not* operated in an irrelevant part of the parameter space (i.e., that “*emittance can be preserved in any scenario*”)—on the contrary, it is highly relevant for future machines.

Reference 41 argues that there is a universal connection between energy-transfer efficiency and transverse (beam-breakup) instability of the trailing bunch. This is based on a simplified model assuming no ion motion, no energy spread, and single-stage acceleration. Ion motion (which will occur for collider beams) introduces nonlinear focusing, which (if controlled) can break the resonance while preserving emittance [Benedetti *et al.*, Phys. Rev. Accel. Beams 20, 111301 (2017)]. BNS (Balakin, Novokhatsky and Smirnov) damping, while in itself insufficient to fully mitigate the instability (unless the energy spread is very large), is in fact able to significantly damp the amplitude growth. Lastly, when using multiple stages, the instability growth is capped by being separated into many shorter plasma stages which are fully or partially uncorrelated (depending on the interstage optics)—in particular if the particles are able to shift longitudinally (i.e., with an R_{56} between the stages). Including all these effects, our preliminary simulations of 100-GeV-class plasma accelerators (to be published as a separate study) indicate the simplistic predictions in Ref. 41 are overly pessimistic.

As an example, consider a 16-stage plasma accelerator from 2 to 50 GeV (at 10^{15} cm^{-3} plasma density, 1 μm rms driver jitter in every stage). Reference 41 predicts an emittance growth of ~ 50 mm mrad, but PIC simulations show that the initial 10-mm-

Fig. R4: Preliminary full-scale simulation (HiPACE++ and ELEGANT) of a 16-stage plasma linac accelerating a 200-pC bunch from 2 to 50 GeV in 100 m (including ~ 50 m of nonlinear interstage optics with non-zero R_{56}). The energy-transfer efficiency is $\sim 20\%$ from the 2-GeV, 2-nC drivers. The timing and alignment jitter is 10 fs and 1 μm rms at every stage (uncorrelated)—consistent with state-of-the-art stability. The top plot shows the stages (blue boxes) and interstages (blue lines). The density plots show the initial (left panel) and final step (right panel) of the full simulation. The various subplots show the evolution of the accelerated beam parameters between the stages (orange and blue lines represent the x and y planes, respectively; the grey line represents angular momentum).

mmrad emittance only grows by 2 mm mrad (to which the interstage optics also contribute). While this example does not itself demonstrate the feasibility of a collider, it is relevant to hard X-ray FELs and “extreme-beam” applications such as strong-field QED experiments. In summary, it is premature to conclude that transverse instabilities are a problem for applications.

Recommendation:

Publish after revision and re-review.

Specific comments and revision requests:

The paper should be adjusted to remove claims of large or significant energy gain.

This point was discussed above.

The last sentence of the abstract should be removed. The second to last sentence should read “We demonstrate, for the first time, emittance preservation in a high-gradient and high-efficiency PWFA.” The claim of charge preservation is questionable, and losses shown in Figure 2 would not be acceptable in energy boosters for photon science or for particle colliders. The remainder of the manuscript should also remove references to charge preservation.

We have softened the wording in the final sentence of the abstract:

(Line 12): This establishes that PWFAs can accelerate without degradation—an essential step toward energy boosters in photon science and multistage facilities for compact high-energy particle colliders.

We do not agree that charge was not preserved, as argued above. In the manuscript, we are very transparent about this not occurring in every shot (but instead 61% of them), and therefore do not find this to be either wrong or misleading.

A few small comments that can easily be corrected:

1) Reference 27 was published in 1991.

We thank the reviewer for pointing this out: it has been corrected in the updated manuscript.

2) Page 7, bottom left, there is a line about “extended data fig. 8 and 9”, is this supposed to be “supplemental”?

This sentence was indeed overlooked during the replacement of “Extended Data Fig.” (as they were previously known), to the new “Supplementary Fig.” This has now been corrected.

Reviewer #2:

The article by C.A. Lindström et al. presents a detailed study on emittance preservation in a plasma wakefield accelerator. It is well-written, clear, and didactic. The achieved control of the beam is impressive, but the main results seem to me too technical to be published in Nature Communications; they seem more suitable for a specialized audience.

With respect, we disagree. Much of the content is unavoidably technical since it is so difficult to produce an environment in which it is possible to make the measurements described. Although as many of the technicalities as possible are relegated to the Methods section, it is essential to allow the reader to understand exactly how the measurements have been made in order to give confidence in the results. We submit that the result is of general importance and of interest not only for a “*specialized audience*” but also to anyone interested in plasma physics and plasma-acceleration methods, as well as anyone interested in future applications of particle accelerators, across a wide variety of scientific problems, that are enabled by plasma-wakefield acceleration.

Emittance preservation is one of the main challenges of PWFA accelerators, which has been thoroughly studied theoretically, showing a trade-off between efficiency and instability. Several techniques have been proposed to mitigate the emittance growth when it occurs. Here, emittance preservation is demonstrated in a small gain regime where no significant emittance growth is expected, for matched and well-aligned beams. The authors showed their ability to fulfill these conditions. They also measured tolerances for mismatching and misalignment, which is a significant result for specialists, but a general audience learns little: the article demonstrates the preservation of emittance in a case where there is no doubt that it should be preserved.

The reviewer seems to be of the opinion that, because a phenomenon has been studied theoretically, it is unnecessary to experimentally verify the conclusions. It seems to us that such a view is contrary to the scientific method. The reviewer criticises us for using a small energy gain but, as we explain in the amended paper, it is essential to start such an investigation with lower energy gain in order to be able to control the various parameters sufficiently well to make the measurement at all. It is for this reason that no-one else has been able to make such a measurement until now. In short, it is necessary to walk before one can run.

The reviewer concludes by commenting that “*there is no doubt that [emittance] should be preserved*”. Once again, this is a statement based on simplified theoretical models and simulations that require experimental validation—experimentation that we have endeavoured to show was neither trivial nor inevitable. The point of Figs. 3 and 4 is to show exactly how sensitive the system is to misalignment and mismatching, but that nevertheless our experimental setup had the required stability, tunability and measurement sensitivity to locate the ideal operating point—currently beyond what any other plasma-accelerator facility is able to achieve.

We also do not agree that the community learns little. Our result shows that beam quality can indeed be preserved in plasma accelerators with a relevant gradient and plasma density. It indicates those aspects that need particular attention in experiments (i.e., misalignment and matching), and how this can be achieved in practice. It shows that particular technology choices for the generation of plasma (i.e., high-voltage discharges) and electron beams (i.e., superconducting linacs) are able to provide the required stability and tunability—perhaps not exclusively, but nevertheless an important input for planning of future high-gain experiments.

All the open questions on emittance preservation and the potential of beam-driven wakefield acceleration for building a collider remain open. This is highlighted in the discussion at the end of the article that evokes this trade-off between efficiency and instability, stresses the importance of the beam breakup instability, and methods for mitigating this phenomenon. Demonstrating one of these solutions would constitute a crucial step toward compact, high-energy particle accelerators for applications such as high-brightness FELs or high-luminosity linear colliders, but this article does not.

It seems that the reviewer is criticising a paper that we have not attempted to write. We nowhere claim to have solved any of the major issues that must be answered to build a collider. What we have done is made a vital first step towards an essential building block, viz. emittance preservation in plasma acceleration. As we address in the discussion section of the amended paper, we do not see the beam-breakup instability in this experiment, it is not expected to contribute to emittance growth an extended simulation (i.e., for an FEL energy booster; see Supplementary Figs. 11–13), and it is not yet established that it will be a show stopper in a collider (see above discussion on pages 7–8). The next step in our experimental programme is repeating these investigations in the more operationally challenging environment caused by a longer cell (500 mm), in order to verify whether our simulations are indeed correct. We hope then to continue to make steady progress towards the goal that both we and the reviewer share: applying plasma-wakefield acceleration to the most difficult applications, such as high-brightness FELs and high-luminosity colliders.

Sincerely,

Carl A. Lindstrøm

On behalf of the co-authors

REVIEWERS' COMMENTS

Reviewer #1 (Remarks to the Author):

I appreciate that the effort required to perform the longer simulation was non-trivial. I agree that the manuscript is significantly improved. Publication should proceed.

Reasonable people will disagree on whether this work shows emittance, energy spread and charge preservation simultaneously. I think this work indicates that all three were preserved on some shots. Some of the questions about charge preservation, and more, are noted in the prior referee reports and continue below. However, it is worth noting that these questions arise because the work is so incredibly thorough it allows many "obvious next steps" to be remarked upon. In the case of disagreement, I believe the Authors should be allowed to present their work as they see fit. The questions surround interpretation, not veracity.

As one example, the charge and energy spread were within the 68th percentile range 41% and 62% of the time. What was the total percentage of shots which preserved both charge and energy spread to the 68th percentile?

DESY | Hamburg

Dr. Carl A. Lindstrøm

(Former) Research Fellow, FTX-AST, FH, DESY
carl.a.lindstroem@desy.de

(Current) Researcher, Dept. of Physics, University of Oslo
c.a.lindstrom@fys.uio.no

27 June 2024

Response to the Reviewers

We would again like to thank the reviewers for their work, and for their positive response. Quoted below are the reviewers' comments (in blue) and our response to each of them (in black).

Reviewer #1:

I appreciate that the effort required to perform the longer simulation was non-trivial. I agree that the manuscript is significantly improved. Publication should proceed.

Reasonable people will disagree on whether this work shows emittance, energy spread and charge preservation simultaneously. I think this work indicates that all three were preserved on some shots. Some of the questions about charge preservation, and more, are noted in the prior referee reports and continue below. However, it is worth noting that these questions arise because the work is so incredibly thorough it allows many "obvious next steps" to be remarked upon. In the case of disagreement, I believe the Authors should be allowed to present their work as they see fit. The questions surround interpretation, not veracity.

As one example, the charge and energy spread were within the 68th percentile range 41% and 62% of the time. What was the total percentage of shots which preserved both charge and energy spread to the 68th percentile?

We appreciate the reviewer's assessment that the manuscript is improved, as well as the acknowledgment that while many more questions could in principle have been discussed, it is meaningful to proceed with publication. In light of this approval by the reviewer, we will not attempt to answer the detailed question at this point.

Nevertheless, in order to allow the reviewer and other interested researchers to investigate further their detailed questions by themselves (such as that asked in the final paragraph), we have made the full dataset and all analysis scripts openly available as a Zenodo repository at <https://doi.org/10.5281/zenodo.11967839>.

Sincerely,

Carl A. Lindstrøm

On behalf of the co-authors